# Reassessing the substrate specificities of the major *Staphylococcus aureus* peptidoglycan hydrolases lysostaphin and LytM

Lina Antenucci[1], Salla Virtanen[2], Chandan Thapa[1], Minne Jartti[1†], Ilona Pitkänen[1], Helena Tossavainen[1], Perttu Permi[1,2,3]*

[1]Department of Biological and Environmental Science, Nanoscience Center, University of Jyvaskyla, Jyväskylä, Finland; [2]Institute of Biotechnology, Helsinki Institute of Life Science, University of Helsinki, Helsinki, Finland; [3]Department of Chemistry, Nanoscience Center, University of Jyvaskyla, Jyväskylä, Finland

*For correspondence:
Perttu.Permi@jyu.fi

Present address: †Faculty of Medicine and Health Technology, Tampere University, Tampere, Finland

### eLife assessment

This manuscript describes a **valuable** study aimed at identifying the substrate specificity of two cell wall hydrolases LSS and LytM in *S. aureus*. The authors show that LytM has a novel function of cleaving D-Ala-Gly instead of only Gly-Gly by using synthetic substrates and **compelling** NMR-based real-time kinetics measurements.

**Abstract** Orchestrated action of peptidoglycan (PG) synthetases and hydrolases is vital for bacterial growth and viability. Although the function of several PG synthetases and hydrolases is well understood, the function, regulation, and mechanism of action of PG hydrolases characterised as lysostaphin-like endopeptidases have remained elusive. Many of these M23 family members can hydrolyse glycyl-glycine peptide bonds and show lytic activity against *Staphylococcus aureus* whose PG contains a pentaglycine bridge, but their exact substrate specificity and hydrolysed bonds are still vaguely determined. In this work, we have employed NMR spectroscopy to study both the substrate specificity and the bond cleavage of the bactericide lysostaphin and the *S. aureus* PG hydrolase LytM. Yet, we provide substrate-level evidence for the functional role of these enzymes. Indeed, our results show that the substrate specificities of these structurally highly homologous enzymes are similar, but unlike observed earlier both LytM and lysostaphin prefer the D-Ala-Gly cross-linked part of mature peptidoglycan. However, we show that while lysostaphin is genuinely a glycyl-glycine hydrolase, LytM can also act as a D-alanyl-glycine endopeptidase.

## Introduction

In the age of antibiotic resistance, multi-resistant bacteria pose a serious threat to global health. This calls for novel strategies to fight the infections (*Shrivastava et al., 2018*). Contemporary means to treat bacterial infections rely on antibiotics. One of the most common mechanisms of action of antibiotics is inhibition of the peptidoglycan (PG) synthesis, a major macromolecular structure in the bacterial cell wall (*Džidić et al., 2008*; *Kapoor et al., 2017*). Alarmingly, the Gram-positive bacterium *Staphylococcus aureus* (*S. aureus*) is notorious in developing resistance towards β-lactam-based antibiotics, e.g., penicillin and their derivatives, which target penicillin binding proteins that are vital for PG synthesis (*Sauvage et al., 2008*). These methicillin-resistant strains of *S. aureus* (MRSA) can cause

**Figure 1.** Structure of the cell wall peptidoglycan (PG) in *S. aureus*. (**A**) Schematic overview of the cell wall in *S. aureus*. (**B**) Structure of the PG. (**C**) Schematic presentation of peptides used to study target bond specificity of the enzymes. This figure was created using BioRender.com.

The online version of this article includes the following figure supplement(s) for figure 1:

**Figure supplement 1.** Overlaid structures of the catalytic domains of lysostaphin (LSS) (green, PDB ID 5NMY) and LytM (magenta, PDB ID 2B13) as well as their aligned amino acid sequences.

life-threatening infections which are very difficult to eradicate (*Lee et al., 2018*). Moreover, although still rarely occurring, outbreaks of infections caused by vancomycin intermediate-level and fully resistant *S. aureus* are lurking on the horizon (*Shariati et al., 2020*). Development of alternative means to treat multi-resistant bacterial infections is therefore needed.

Cell division, cell shape determination, PG remodelling, and recycling are coordinated and executed by peptidoglycan hydrolases (PGHs), enzymes produced to function as (auto)lysins in the regulation of the cell wall during growth and division (*Ghuysen, 1968*; *Wang et al., 2022*). On the other hand, PGHs may also function as defensive weapons against other bacterial species as in the case of lysostaphin (LSS), an exolysin secreted by *Staphylococcus simulans* biovar staphylolyticus, which displays bactericidal action against competing *S. aureus* (*Schindler and Schuhardt, 1964*). The potential antibacterial role of PGHs is based on the targeted destruction of the protecting PG by hydrolysis, which leads to lysis of the bacterial cells upon increasing turgor pressure (*Ghuysen, 1968*; *Schindler and Schuhardt, 1964*). Given that PGHs are promising bactericins as well as druggable targets for the treatment of multidrug-resistant *S. aureus* infections, profound knowledge of their structure, function, as well as substrate specificity is instrumental to harness their full potential as a new breed of antibiotics (*Szweda et al., 2012*).

*S. aureus* and other Gram-positive bacteria are protected by a thick PG layer that is composed of repeating β-1,4-linked *N*-acetylmuramic acid (MurNAc) and *N*-acetylglucosamine disaccharide units, forming the conserved glycan backbone of murein (*Ghuysen, 1968*; *Schleifer and Kandler, 1972*; *Figure 1A, B*). Each MurNAc carboxyl group is linked to a stem peptide

(L-Ala-D-iso-Gln-L-Lys-D-Ala-D-Ala) and two stem peptides are connected via a cross-bridge structure, the exact composition, and length of which depends on the bacterial species in question. In *S. aureus* the cross-bridge is characteristically composed of five glycine residues (*Ghuysen, 1968*). Cross-bridging provides an integral structural support for the entire PG wall and allows PG to reach a thickness of over 40 layers (20–80 nm) in Gram-positive bacterial species (*Ghuysen, 1968*; *Kim et al., 2015*).

While cell wall composition and biosynthesis of PG are relatively well understood, the perception of PG maintenance and hydrolysis at the structural level is more vague (*Szweda et al., 2012*). One prominent PGH family are the LSS-like endopeptidases, which specifically target the cross-bridge peptide bonds of *S. aureus* PG (*Ghuysen et al., 1966*; *Vollmer et al., 2008*). Here, we compared LytM and LSS, which both belong to the zinc-dependent M23 family of metalloendopeptidases and are designated as glycyl-glycine hydrolases. They have been shown to cleave the peptide bonds between glycine residues in the pentaglycine cross-bridge of *S. aureus* PG (*Figure 1B*; *Bardelang et al., 2009*; *Browder et al., 1965*; *Iversen and Grov, 1973*; *Odintsov et al., 2004*; *Ramadurai et al., 1999*; *Raulinaitis et al., 2017*; *Schindler and Schuhardt, 1964*; *Schindler and Schuhardt, 1964*; *Tossavainen et al., 2018*; *Warfield et al., 2006*).

*S. simulans* LSS is an exolysin possessing bacteriolytic properties towards staphylococci with a pentaglycine cross-bridge structure, e.g., *S. aureus*, *S. carnosus*, *S. cohnii* (*Grabowska et al., 2015*; *Schindler and Schuhardt, 1964*; *Schleifer and Fischer, 1982*). LytM is a *S. aureus* PG hydrolase with a catalytic M23 domain structurally very similar to that of LSS (*Figure 1—figure supplement 1*). The common fold consists of a characteristic narrow groove formed by a β sheet and four surrounding loops. At one end of the groove resides the catalytic site in which a zinc cation is coordinated by two conserved histidines and an aspartate. The zinc cation, which polarises the peptide bond, and a nucleophilic water molecule activated by two other conserved histidines act in concert to hydrolyse the substrate glycyl-glycine bond (*Grabowska et al., 2015*).

Despite tens of years of effort in studying substrate specificities of LSS-like M23 endopeptidase family members, the exact molecular targets of these enzymes have remained enigmatic, contradictory, and yet imprecise. Interpretation of results is further complicated by diverse conventions used for the numbering of cross-bridge residues. Indeed, the exact cross-bridge bond that LSS targets remains controversial; Browder et al. reported LSS cleavage between glycines 4 and 5 while Sloan and colleagues observed cleavage of all glycyl-glycine bonds with several substrates of different sizes (*Browder et al., 1965*; *Sloan et al., 1977*). Using mass spectrometry (MS), Schneewind and colleagues observed LSS-mediated cleavage between glycines 3 and 4 (*Schneewind et al., 1995*). This observation was supported by a recent NMR spectroscopic study highlighting that hydrolysis by LSS produced fragments in which either two or three glycines are interlinked to the lysine sidechain (*Maya-Martinez et al., 2018*). On the other hand, Xu and coworkers reported cleavage at several sites with digested muropeptides: between glycines 1 and 2, 2 and 3, as well as 3 and 4 (*Xu et al., 1997*). Moreover, studies carried out with FRET peptides reported cleavage between glycines 2 and 3 (~60 %) and glycines 3 and 4 (~40%) (*Warfield et al., 2006*).

Much less is understood about the functional role of *S. aureus* PG hydrolase LytM. Based on its high sequence homology with LSS catalytic domain as well as its association with *S. aureus* cell wall maintenance, LytM has implicitly been envisaged to target pGly cross-bridges in the *S. aureus* cell wall. LytM has been shown to hydrolyse pGly into di- and triglycine, and tetraglycine into diglycine (*Firczuk et al., 2005*). Understanding the substrate specificity is of utmost importance for deciphering the functional role of endogenous LytM in cell division, PG remodelling, antimicrobial resistance, and PG synthesis/hydrolysis in general.

Thus far, specificity of PG hydrolases has been determined utilising end-point kinetics together with LC/MS analysis of digested PG fragments or turbidity reduction assays to measure the site(s) and efficiency of hydrolysis (*Frankel et al., 2011*; *Małecki et al., 2021*). However, these approaches have several pitfalls, e.g., they do not provide information on reaction kinetics or atomic resolution of reaction products, which severely limits determination of substrate specificity. In seeking to understand the functional role of these enzymes we have deciphered substrate specificity of LSS and LytM catalytic domains through a systematic approach by monitoring hydrolysis of various synthetised PG fragments as well as muropeptides from purified sacculus extracted from *S. aureus* cells using solution-state NMR spectroscopy and by utilising turbidity reduction assay on living *S. aureus* cells.

Most importantly, NMR enables following the hydrolysis reaction in real time as well as identification of hydrolysis products.

Here, we show that the substrate specificities as well as catalytic efficiencies of M23 family PGHs targeting *S. aureus* PG are different from what has been earlier anticipated. Moreover, our results reveal that quite unexpectedly LytM substrate specificity extends beyond glycyl-glycine endopeptidase activity, which calls for revision of its classification.

## Results

### Substrate specificities of LSS M23 enzyme family

Due to inconsistent nomenclature used in the literature to annotate *S. aureus* PG cross-bridge structure and hence substrates used in the enzymatic assays and specificity studies, we conducted a systematic approach by dissecting the complex structure of PG using synthetic peptides spanning from the simplest pGly to more complex branched peptides to elucidate the substrate specificity of LSS and LytM (*Figure 1C*, *Supplementary file 1*). The scaffold structure in the PG fragments used in this study is the pGly cross-bridge as it is uniquely present in *S. aureus* PG (*Figure 1B*).

*Figure 2* displays the strategy employed to study substrate specificity and kinetics of LSS and LytM using two different approaches: (i) monitoring hydrolysis of synthetic peptides mimicking PG fragments together with muropeptides extracted from *S. aureus* sacculus using solution-state NMR spectroscopy, and (ii) monitoring lysis of bacterial cell wall using turbidity reduction assay.

As pGly is recognised as the common physiological substrate for M23 family endopeptidases, we wanted to accurately define substrate specificities, catalytic efficiencies, and the sites of cutting for the catalytic domains of LSS and LytM. To measure the rate of pGly hydrolysis in vitro, we added the enzyme and monitored decaying substrate concentration with respect to reaction time using quantitative $^1$H NMR spectroscopy (*Figure 2A*). Results indicate that LSS hydrolyses pGly 15-fold faster than LytM in vitro (*Figure 2B*). For comparison, we measured the outcome of hydrolysis of extracted muropeptides using end-point kinetics (*Figure 2C*, vide infra). In the turbidity reduction assay, lytic efficiencies of externally administered LSS and LytM against *S. aureus* USA300 (MRSA) strain were compared (*Figure 2D*). These data show that $OD_{600}$ of late stationary cells reduced from 100% to 25% in 1.5 and 12 hr for LSS and LytM, respectively. Thus, LSS and LytM display consistent differences in efficiencies in the turbidity and pGly assays. However, *S. aureus* cell wall as a macroscopic substrate or cellular milieu differs significantly from the conditions used for the kinetic assay in vitro, i.e., pGly as a substrate represents a poor model for describing substrate specificity and functional differences of M23 family endopeptidases. As the next step, we therefore determined the scissile bonds in pGly as well as extended substrate specificity studies of these enzymes beyond the pGly cross-bridge.

### LSS and LytM preferentially hydrolyse the $Gly_2$-$Gly_3$ bond in pentaglycine

We recently showed using a two-dimensional (2D) $^{13}$C-HMBC NMR experiment that LytU hydrolyses pGly into di- and triglycine (*Raulinaitis et al., 2017*). Whether this was the result of hydrolysis of the bond between $Gly_2$ and $Gly_3$, and/or between $Gly_3$ and $Gly_4$ in pGly remained undefined because the products are the same in both reactions. Selective $^{15}$N,$^{13}$C-labelling (*Figure 2E*) breaks the isotopic symmetry of pGly without introducing per se any non-physiological tags to the substrate and enables to define the preferred cleavage sites for LSS and LytM in pGly using $^1$H-$^{13}$C NMR spectroscopy optimised for glycine detection (*Figure 2F* and *Figure 2—figure supplement 1*). As can be appreciated in *Figure 2G and H*, isotopic labelling of $Gly_2$ at Cα and CO carbons in pGly allows unambiguous determination of cleavage site, because the characteristic chemical shift of a C-terminal $^{13}$CO resonance (179.4 ppm) is markedly different from a non-terminal $^{13}$CO chemical shift (173.8 ppm) at physiological pH. Representative 2D $^1$Hα-$^{13}$CO correlation maps, collected with the glycine-optimised 2D HA(CA)CO NMR experiment (*Figure 2F* and *Figure 2—figure supplement 1*) from the selectively $Gly_2$ $^{13}$C-labelled pGly, are highlighted in *Figure 2H*. These data clearly show that LytM and LSS are highly specific for the bond between $Gly_2$ and $Gly_3$ (>85–94%, *Figure 2I*). However, residual cleavage activity towards the bond between $Gly_3$ and $Gly_4$ is also evident (<15%).

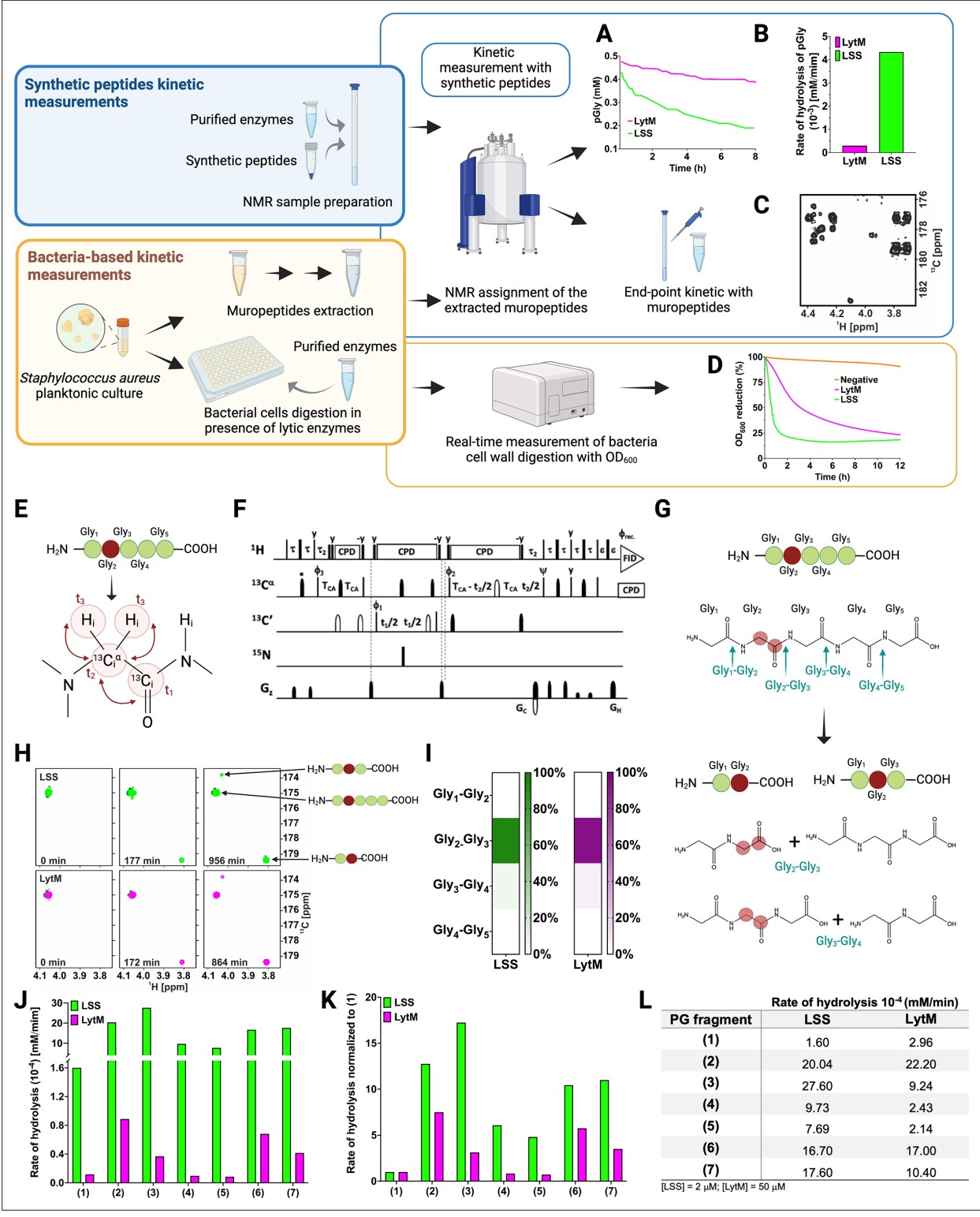

**Figure 2.** Workflow to study M23 peptidoglycan hydrolase (PGH) substrate specificities. Panels in the upper left corner, the two main strategies used in the study. Kinetic measurements carried out with peptidoglycan (PG) fragments (synthetic peptides) were supported by bacteria-based kinetic measurements using *S. aureus* USA300 cells. Panels in the upper right corner, (**A**) hydrolysis of synthetic pGly (mM) by lysostaphin (LSS) (green) and LytM (magenta) monitored by $^1$H NMR spectroscopy over time (hr). Both enzymes were used at the concentration of 50 µM. (**B**) Rate of hydrolysis (mM/min) of pGly derived from A in the first 60 min of the reaction for LSS and LytM. (**C**) $^{13}$C-HMBC NMR spectrum showing the end-point kinetic of LSS-treated muropeptides extracted from *S. aureus* USA300 cells. (**D**) Turbidity assay using *S. aureus* USA300 cells in the presence of LSS and LytM

*Figure 2 continued on next page*

*Figure 2 continued*

at a concentration of 3 µM. The cell lysis is expressed as percentage reduction of the bacteria suspension optical density at 600 nm over time (hr). (**E**) Pentaglycine hydrolysis by LSS and LytM was studied by using a Gly$_2$$^{13}$C-labelled substrate. (**F**) NMR pulse sequence for the acquisition of glycine Hα-detection optimised two-dimensional (2D) HA(CA)CO spectra, showing correlations between $^1$Hα and $^{13}$CO atoms. (**G**) With a label the otherwise identical products of the two hydrolysis reactions G$_2$-G$_3$, G$_3$-G$_4$ can now be differentiated. (**H**) The order of appearance of the peaks of the labelled products as a function of time. The labelled glycine has a different $^{13}$CO shift when as G$_2$ in triglycine or G$_2$ in diglycine. (**I**) Heatmap summarising the bond preferences for the enzymes in the pGly. Hydrolysis of peptides 1–7 by LSS (green) and LytM (magenta). (**J**) Initial rates of substrate hydrolysis (mM/min) of LSS and LytM at 2 µM concentration and (**K**) the same rates normalised to that of pGly. (**L**) Absolute values of rates of hydrolysis. For PG fragments **2** and **3**, two independent measurements were performed to test and accredit the reproducibility of the method (see Materials and methods). This figure was created using BioRender.com.

The online version of this article includes the following source data and figure supplement(s) for figure 2:

**Source data 1.** Raw Excel data for *Figure 2D*.

**Figure supplement 1.** Glycine-optimised HCACO experiment for correlating $^1$Ha, $^{13}$Ca, $^{13}$CO resonances.

**Figure supplement 2.** Hydrolysis of peptides 1–7 by lysostaphin (LSS) (green) and LytM (magenta).

**Figure supplement 3.** Assessment of the stability of the 800 MHz $^1$H NMR spectrometer during the relatively long periods of acquisition of kinetics spectra.

## The substrate specificity of LytM and LSS is determined by the D-Ala-Gly cross-link between adjacent PG monomers

In the next phase, we sought to understand the role of plausible auxiliary contacts arising from stem peptides flanking the pGly cross-bridge for the substrate specificity of LSS and LytM. To this end, several larger peptides with PG-specific extensions around the pGly scaffold were utilised (*Figure 1C*). To alleviate complications arising from multiple sites of hydrolysis in pGly, we decided to monitor the rate of substrate hydrolysis rather than the rate of product formation, and to use a similar substrate concentration for all the PG fragments (~0.4 mM). These data using substrates **1–7** are shown in *Figure 2—figure supplement 2* (see also *Figure 2—figure supplement 3*). Normalisation of the absolute rates with respect to the common scaffold structure, (pGly) should reveal on the substrate level the preference of LSS and LytM towards different PG fragments (*Figure 2K*).

Clearly, the overall rate of hydrolysis for LytM and LSS increases with peptides that mimic cross-linked PG fragments (*Figure 2J*). Indeed, cross-linking L-Lys-D-Ala dipeptide N-terminally to pGly in **2** increased the rate of hydrolysis with respect to **1** by a factor of 12 and 8 for LSS and LytM, respectively (*Figure 2K*). Further elongation in the N-terminus (ADiQKDA-GGGGG, **3**) had only incremental contribution to the rate of overall substrate hydrolysis, **17** and **3** for LSS and LytM, respectively.

Next, we inspected the influence of a stem peptide linked to the C-terminus of pGly on the substrate hydrolysis rate. In PG fragment **4** the stem peptide is linked to the pGly via an isopeptide bond between lysine ε amino group and the C-terminus of pGly. The PG fragment **5** is similar but corresponds to the peptide moiety in a PG monomer (or pentaglycyl-Lipid II) (*Figure 1C*). For LSS, the rate of hydrolysis increased by six- and fivefold for substrates **4** and **5**, respectively. This increase in overall rate of substrate hydrolysis with respect to pGly is roughly two and three times smaller than for the linear PG fragments **2** and **3** that contain a D-Ala-Gly cross-link, respectively. For LytM, the outcome was in stark contrast to the results observed with LSS. Indeed, LytM activity towards PG fragments **4** and **5** is only 82% and 72%, respectively, of that found for **1**.

Next, we studied the significance of capping the pGly moiety from both its termini. PG fragments **6** and **7** both contain an N-terminal D-Ala-Gly cross-linkage in their structure but differ in **6** having a C-terminal tetra- and **7** a pentapeptide stem, i.e., the latter has a D-Ala-D-Ala moiety in its structure. For LSS, 10-fold rate enhancements with respect to **1** for substrate hydrolysis were observed, comparable to enhancements with substrates **2** and **3,** indicating that the moiety C-terminal to pGly does not contribute to LSS catalytic efficiency. Hence, based on these results, it can be deduced that LSS strongly favours substrates that contain a D-Ala-Gly cross-link, i.e., **2**, **3**, **6**, and **7**. In the case of LytM, rate enhancements of PG fragments **6** and **7** with respect to **1** were 5.5- and 3.5-fold, respectively, meaning that the branched stem peptide linked to the C-terminal end of pGly has a slight negative effect on LytM activity. These data indicate that similarly to LSS, LytM is more specific towards substrates **2**, **3**, **6**, and **7**, i.e., PG fragments that contain a D-Ala-Gly cross-link.

In all, overall rates of substrate hydrolysis using comparative real-time kinetics convincingly indicate that LytM as well as LSS prefer mature PG fragments as substrates i.e., PG units that contain a D-Ala-Gly cross-link between two stem peptides. In contrast to the mere end-point kinetics carried out in the past for LytM and LSS, our present results clearly demonstrate the preference of these enzymes for cross-linked PG fragments beyond redundant glycyl-glycine endopeptidase activity. However, the substrate-specific real-time kinetics assay does not tell anything about the site(s) of hydrolysis in the substrates. To understand substrate specificity and the physiological role of these PGHs, we inspected the outcome of these reaction assays in more detail.

## LSS and LytM display different substrate specificities

NMR spectroscopy allows the identification of atom connectivities within the substrate and products, enabling determination of site(s) of hydrolysis at atomic resolution. *Figure 3A and B* display the LSS and LytM, respectively, catalysed hydrolysis of substrate **2**, the simplest of our PG fragments containing the cross-link between D-Ala and $Gly_1$ of pGly synthesised by the transpeptidase. For LSS, expansion of the region corresponding to the $^1H\alpha$ resonance of D-Ala in the $^1H$ spectrum of **2** shows that upon hydrolysis of **2**, two products are formed as manifested themselves by increasing concentrations of $KDAG_1$ and $KDAG_1G_2$ (*Figure 3A*, *Figure 3—figure supplement 1* shows the identification of the product peaks based on a $^{13}C$-HMBC spectrum). Owing to their different chemical structures, the products display separate peaks, resolved at 800 MHz $^1H$ field, which allows determination of individual reaction rates. Given that the $^1H$ method we used is quantitative the preferred site of hydrolysis in the PG cross-bridge can be identified based on product concentrations (*Figure 3C and E*). These data show that in **2** LSS hydrolyses amide bonds between $Gly_1$-$Gly_2$ and $Gly_2$-$Gly_3$ with a clear preference for the first one.

By coupling real-time kinetics data with the identification of reaction products, we determined the cutting sites and the reaction rates of the corresponding products for a total of seven PG fragments for LSS. The combined results of their analyses are shown in *Figure 3E*. The difference in panels C and E is that for the latter we considered only reactions which dominate for the particular PG fragment whereas in the heatmap representation the relative product concentrations at the end of the hydrolysis reaction are shown. These results are consistent with analyses of substrate hydrolysis rates by showing that LSS clearly favours PG fragments with a D-Ala-Gly cross-link. Pentaglycine or other PG monomers in which $Gly_1$ is not cross-linked to the stem peptide of the adjacent PG fragment are hydrolysed at slower rate and the cutting site is shifted by one residue downstream compared to the cross-linked ones. In addition, the bond specificity of the hydrolysis reaction increases with D-Ala-Gly cross-linked PG fragments. In this case, LSS favours cutting the amide bond between $Gly_1$ and $Gly_2$ and the hydrolysis rate of this bond is several times higher than that of the $Gly_2$-$Gly_3$ bond, or those of $Gly_2$-$Gly_3$ and $Gly_3$-$Gly_4$ bonds in PG monomers devoid of D-Ala-Gly cross-link.

As we reckoned that D-Ala-Gly cross-link is critical for efficient hydrolysis of LSS substrates, we wanted to study how the length of the glycine containing cross-bridge influences substrate hydrolysis. Substrate **12**, containing four consecutive glycines attached to Lys-D-Ala, was hydrolysed between $Gly_1$ and $Gly_2$, as well as $Gly_2$ and $Gly_3$, exactly like substrate **2** (*Figure 3—figure supplements 2–3*). However, the hydrolysis rate of the $Gly_2$-$Gly_3$ bond was threefold faster than in the case of **2**. This is probably due to the absence of tetraglycine as a secondary substrate in the reaction, i.e., the reaction products resulting from hydrolysis of **12** do not significantly compete with the primary substrate for binding to the LSS catalytic centre. Substrate **13** (KDAGGG) was hydrolysed very slowly and **14** (KDAGG) not at all.

We carried out a similar analysis for LytM using the same set of PG fragments (*Figure 3B, D, and F*). The most striking observation is clearly visible in *Figure 3B* showing hydrolysis of **2** by LytM, which results in formation of products KDA and $KDAG_1$. Indeed, given that LytM is a well-established glycyl-glycine endopeptidase (*Firczuk et al., 2005*; *Grabowska et al., 2015*; *Ramadurai et al., 1999*), we were taken by surprise to observe that LytM, in addition to $Gly_1$-$Gly_2$ bond hydrolysis, is also able to cleave the D-Ala-$Gly_1$ amide bond in the *S. aureus* PG cross-bridge whenever the cross-linked D-Ala-Gly structure is available.

We verified that the observed D-alanyl-glycine bond hydrolysis, which has not been reported before this study, is stereospecific and linked to LytM activity. We performed an identical kinetic assay using KLAGGGGG as a negative control. It confirmed that the enzyme is indeed D-enantiomer

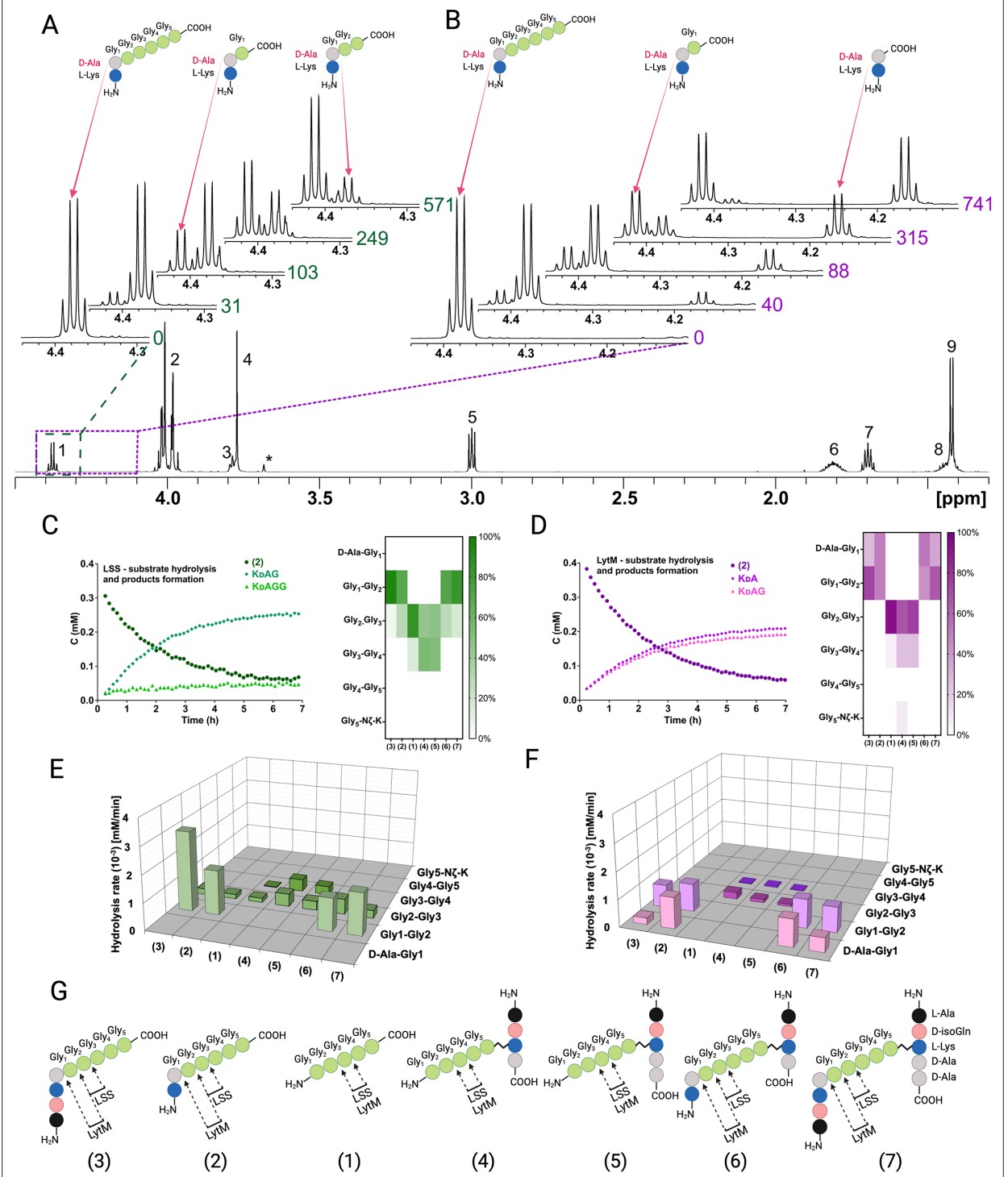

**Figure 3.** Representative examples of real-time NMR monitoring of substrate hydrolysis. Quantitative ¹H spectra at selected time points in the hydrolysis reactions of peptide **2** by lysostaphin (LSS) (**A**) and LytM (**B**). In hydrolysis by LSS peaks of Ala Hα in products KDAG and KDAGG gradually appear as a function of time, whereas in LytM reaction KDA and KDAG are formed. Time points are given in minutes next to the spectra. Peak assignments in the reference spectrum are the following: 1 Ala Hα, 2 Gly₁-Gly₄ Hα, 3 Lys Hα, 4 Gly₅ Hα, 5 Lys Hε, 6 Lys Hβ, 7 Lys Hδ, 8 Lys Hγ, 9 Ala Hβ. Asterisk marks the peak of the buffer. The alanine Hα quartets of substrate (4.373 ppm) and KDAGG (4.365 ppm) partially overlap. (**C, D**) On the left, concentrations in function of reaction time derived from NMR peak integrals from a typical reaction setup with 0.4 mM peptide and 2 µM LSS or 50 µM LytM. On the right, relative product concentrations at reaction end points for the studied peptidoglycan (PG) fragments. (**E, F**) Rates of formations of products in

*Figure 3 continued on next page*

*Figure 3 continued*

hydrolyses by LSS and LytM of the studied PG fragments **1–7**. (**G**) Bonds cleaved by LSS and LytM in different PG fragments. This figure was created using BioRender.com.

The online version of this article includes the following source data and figure supplement(s) for figure 3:

**Source data 1.** Raw Excel data for *Figure 3A*.

**Figure supplement 1.** Identification of products in the hydrolysis reaction of peptide **2** by lysostaphin (LSS) with a $^{13}$C-HMBC spectrum.

**Figure supplement 2.** Hydrolysis of peptides KDAGGGG (**12**), KDAGGG (**13**), and KDAGG (**14**) by lysostaphin (LSS) (**A**) and LytM (**B**).

**Figure supplement 3.** Hydrolysis of peptidoglycan (PG) fragments **12–14** by lysostaphin (LSS) and LytM.

**Figure supplement 4.** Hydrolysis of peptide KLAGGGGG (**15**) by LytM.

**Figure supplement 5.** Hydrolysis of muropeptides extracted from *S. aureus* USA300 sacculus by lysostaphin (LSS) and LytM.

specific as no cleavage between alanine and glycine was observed (*Figure 3—figure supplement 4*). Further, using a protocol identical to that used for LytM we engineered, expressed, and purified the inactive H291A mutant of LytM and tested it with substrate **2**. As expected, LytM H291A showed no activity. The results confirm that in addition to being a glycyl-glycine endopeptidase as reported earlier (*Firczuk et al., 2005*; *Grabowska et al., 2015*; *Ramadurai et al., 1999*), LytM belongs to a small family of endopeptidases which cleave the peptide bond between D-Ala and Gly.

In substrates **2** and **6** LytM cleaves between D-Ala-Gly and $Gly_1$-$Gly_2$ with approximately equal rates. Using substrate **2** we did not observe a significant concentration-dependent change in the relative reaction rates of D-Ala-Gly and $Gly_1$-$Gly_2$ hydrolysis in the [S]/[E] range tested, from 725:1 to 1:2, when [LytM]=50 µM. For the N-terminally longer substrates **3** and **7**, the balance is shifted in favour of hydrolysis of $Gly_1$-$Gly_2$. Regarding substrates lacking the D-Ala-Gly cross-link, LytM seems to disfavour the stem peptide in the C-terminus of pGly and the overall rate of hydrolysis with respect to pGly is diminished more drastically in comparison to LSS.

Analogously to LSS, we compared the rate of hydrolysis of shorter PG fragments with LytM to map the critical length of the glycine-bridge in the substrate needed for the hydrolysis (*Figure 3—figure supplements 2–3*). Compared to **2**, the rate of hydrolysis did not change drastically when shortening the glycine chain from five glycines to three. However, LytM strongly favoured D-Ala-Gly cleavage in **13** (KDAGGG) whereas it preferred $Gly_1$-$Gly_2$ cleavage in **12** (KDAGGGG). Unlike LSS, LytM can still hydrolyse **14** (KDAGG). Only the D-alanyl-glycine bond is cleaved, and the reaction is five to ten times slower than hydrolysis of **2**, **12**, and **13**. Owing to its D-Ala-Gly cleavage efficiency, we tested whether LytM could also hydrolyse KDAAGGGGG and KDAAA type substrates. However, the former was hydrolysed like pGly, i.e., no D-Ala-L-Ala cleavage occurred, but KDAAGG and KDAAGGG products were observed instead (not shown). *Enterococci* cell wall PG mimicking peptide KDAAA was not cleaved at all. This data further confirms that LytM is not only a glycyl-glycine endopeptidase but also acts as a D-alanyl-glycine hydrolase.

To confirm the scissile bond specificity determined for LSS and LytM using synthetic PG fragment mimicking peptides, we extracted and purified muropeptides from *S. aureus* USA300 sacculus using the established protocol (*Kühner et al., 2014*). After administering muropeptide samples with LSS and LytM, we observed hydrolysis of the same amide bonds as with corresponding synthetic PG fragments (*Figure 1C*, *Figure 3—figure supplement 5*). LSS hydrolyses the peptide bond between $Gly_1$ and $Gly_2$, recognised as the appearance of interconnected C-terminal $Gly_1$ $^1H\alpha$-$^{13}CO$ and D-Ala $^1H\alpha$-$^{13}CO$ peaks in the $^{13}$C-HMBC spectrum (*Figure 3—figure supplement 5*). Identical correlations are observed for LytM, indicating that it also hydrolyses the amide bond between $Gly_1$ and $Gly_2$ (*Figure 3—figure supplement 5*). However, additional resonances stemming from the C-terminal D-Ala that is correlated with the neighbouring Lys are also observed. This indicates that LytM is cutting both glycyl-glycine and D-alanyl-glycine bonds also in muropeptides extracted from *S. aureus* sacculus.

## Differences in susceptibilities of *S. aureus* mutants towards LSS and LytM can be explained by their substrate specificities

The composition of the cell wall PG of *Staphylococci*, including, e.g., *S. aureus*, *S. simulans*, *S. epidermidis*, can be modulated by intracellular enzymes. For instance, gene products of *fem* (factor essential to methicillin resistance) family members, i.e., FemX, FemA, and FemB catalyse nonribosomal

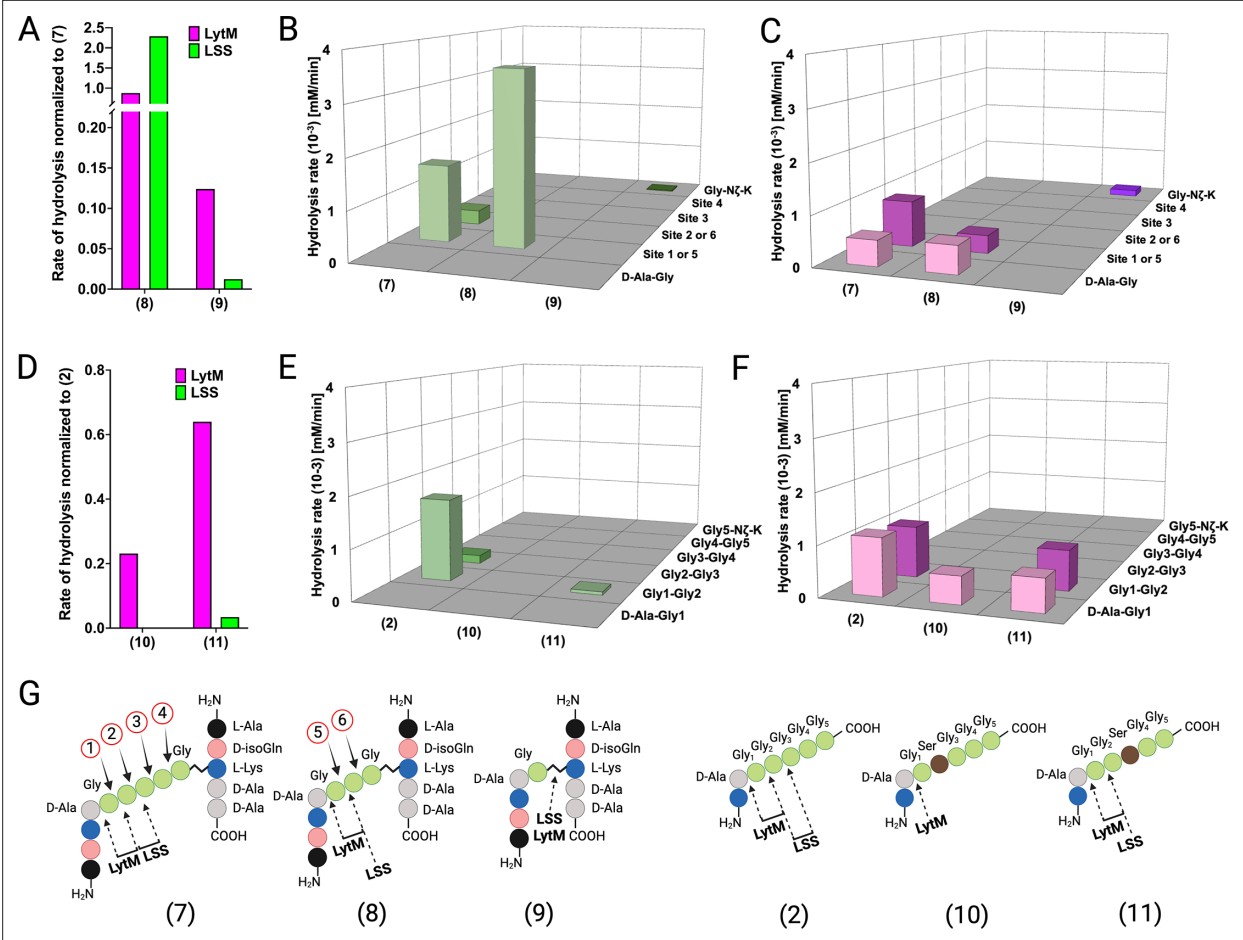

**Figure 4.** Hydrolysis of peptidoglycan (PG) fragments with a shorter cross-bridge or with serine in cross-bridge. Rates of substrate hydrolysis of fragments **8** and **9** as compared with **7** by lysostaphin (LSS) (green) and LytM (magenta) (**A**) and formation of product(s) in hydrolysis by LSS (**B**) and LytM (**C**). Rates of substrate hydrolysis of fragments **10** and **11** as compared with **2** (**D**) and formation of product(s) in hydrolysis by LSS (**E**) and LytM (**F**). Depictions of structures of used PG fragments (**G**). LytM was used at a concentration of 50 μM while LSS was 2 μM. This figure was created using BioRender.com.

insertion of mono-(Gly$_5$), di-(Gly$_4$-Gly$_3$), and di-(Gly$_2$-Gly$_1$) glycines into the glycine cross-bridge in *S. aureus* PG, respectively (**Gründling et al., 2006**). Reduced susceptibility of *S. aureus* Δ*femB* and Δ*femAB* mutant strains towards LSS has been observed earlier (**Małecki et al., 2021**; **Strandén et al., 1997**). We pondered whether the length of the cross-bridge of *S. aureus* could influence the catalytic efficiency or target bond specificity of LSS and LytM. To this end, we studied hydrolysis of PG fragments **8** and **9** mimicking tri- and monoGly cross-bridge composition of PG in *S. aureus* Δ*femB* and Δ*femAB* mutants, respectively (**Figure 4**). Interestingly, LSS hydrolyses **8** with an increased rate as compared to **7** (**Figure 4A**). Furthermore, specificity increases, i.e., cutting of glycyl-glycine bond corresponding to the site '2' in **7** and site '6' in **8** is not observed (**Figure 4B**). Quite surprisingly, we also observed that LSS is able to hydrolyse substrate **9**, devoid of glycyl-glycine bond. Instead, LSS hydrolyses the Lys N $\zeta$ -monoGly CO isopeptide bond. The rate of hydrolysis is, however, drastically lower (1.2%) than that of the amide bond between glycines in **7** (**Figure 4A and B**). This result is in excellent agreement with earlier studies on *S. aureus* Δ*femB* and Δ*femAB* mutants and muropeptides extracted from these strains. Indeed, lytic efficiency of LSS against Δ*femB* mutant is not drastically reduced while the minimum inhibitory concentration is three orders of magnitude higher for the Δ*femAB* mutant (**Gründling et al., 2006**). A dramatic reduction of LSS lytic performance against Δ*femAB* mutants was likewise observed in a turbidity reduction assay (**Małecki et al., 2021**). However, residual lytic activity of LSS can be explained by its ability to hydrolyse the Lys-monoGly isopeptide bond existing in the Δ*femAB* mutant.

Similar results were obtained with LytM except for the scissile bond specificity. Even if the number of glycines in the cross-bridge is reduced from five to three, LytM is still able to hydrolyse both D-Ala-Gly and glycyl-glycine bonds (sites '1' or '5') (*Figure 4A and C*). This is in accordance with results on shortened linear peptides (*Figure 3—figure supplement 3*) as well as with previous turbidity reduction assay data on ΔfemB mutant (*Małecki et al., 2021*). However, if stem peptides are cross-linked with only a single glycine in the cross-bridge (**9**), LytM displays diminished activity by eightfold in comparison to **7**. Yet somewhat surprisingly, considering the cleavage of D-Ala-Gly bond in substrate **14** (KDAGG, *Figure 3—figure supplement 3*), LytM hydrolysed the Lys N$\zeta$-monoGly CO isopeptide bond. This again provides rationale for non-negligible lytic activity observed for LytM on ΔfemAB mutants although devoid of glycyl-glycine peptide bonds in their cross-bridge (*Małecki et al., 2021*).

Next we tested the activity of LSS and LytM on PG fragments originating from the so-called LSS immunity factor (Lif/epr)-containing strains of *S. aureus*, i.e., the significance of serine substitutions in the PG cross-bridge on substrate hydrolysis (*Sugai et al., 1997*; *Thumm and Götz, 1997*; *Tschierske et al., 2006*). Of the two PG fragments **10** and **11**, having $Gly_2$ or $Gly_3$ replaced by a serine, only **11** was hydrolysed by LSS. LSS cleaved the bond between $Gly_1$ and $Gly_2$ in **11** although with a rate 30-fold slower than that for substrate **2** (*Figure 4D and E*). The results with LSS correlate well with earlier observations that show only fractional lytic activity of LSS towards staphylococcal strains with serine in the cross-bridge (*Małecki et al., 2021*). LytM was able to hydrolyse both PG fragments. **11** was cleaved similarly to the PG fragment **2**, whereas LytM specifically cleaved the D-Ala-Gly amide bond in **10** as it contains serine in the second position. The rates of substrate hydrolysis were not drastically lower, i.e., by a factor of 4.3 (**10**) and 1.5 (**11**) in comparison to **2** (*Figure 4D and F*). These data provide rationale to the observed differences in lytic efficiencies of LSS and LytM on *S. aureus* cells having serine substitutions in the cross-bridge (*Małecki et al., 2021*). Hence, given that LytM exhibits significant catalytic activity towards D-Ala-Gly cleavage, it is less susceptible to serine substitutions in the cell wall PG.

## Structural basis for the recognition and cleavage of PG by LSS and LytM

We combined the detailed substrate-level information in terms of real-time reaction kinetics and scissile bond specificities together with the existing structural models available for LSS and LytM to glean structural-level understanding of enzyme specificities. Based on these data and using the nomenclature formulated by *Schechter and Berger, 1967*, we delineated substrate specificities for LSS and LytM (*Figure 5*). It is clear that LSS is a glycyl-glycine endopeptidase as P1 and P1' positions are invariably occupied by Gly residues (*Figure 5A*). However, the rate of hydrolysis increases when D-Ala occupies the P2 position, i.e., when LSS recognises a D-Ala-Gly cross-link in the cell wall. LytM is flexible regarding the P1 site, it can be accommodated either by D-Ala or Gly, whereas the P1' position is invariably occupied by Gly (*Figure 5B*).

Why does LytM hydrolyse a D-Ala-Gly bond but LSS does not? To address this intriguing question, we utilised the existing structure of LytM in complex with the transition state analog tetraglycine phosphinate (*Grabowska et al., 2015*) and used molecular modelling approach to dock PG fragment **2** into the active sites of LSS and LytM (*Figure 5C–F*). As has been proposed earlier, the $Zn^{2+}$ at the active site polarises the scissile bond by coordinating the carbonyl oxygen of the residue in the P1 position. For LSS it is Gly, and for LytM it is either D-Ala or Gly (*Figure 5C–F*). D-Ala in the P1/S1 position in the LSS active site results in a steric clash with loop 1, most notably its residues L272-I274, thus preventing hydrolysis of a D-alanyl-glycine peptide bond (*Figure 5E*). In LytM the corresponding loop is shorter allowing accommodation of D-Ala in the P1/S1 position and hence cleavage of D-alanyl-glycine cross-bridge or alternatively, if the substrate has a Gly in the P1 position, the peptide bond between $Gly_1$ and $Gly_2$ (*Figure 5D and F*).

The same loop 1 gates the LSS active site and establishes the structural basis for Lif function. Serine in the P2' position severely hinders accommodation of the substrate to the LSS active site, which results in a drastic drop in the rate of hydrolysis (*Figure 5A and G*). In LytM the three residues shorter loop renders the active site more voluminous, which permits the short polar sidechain of Ser to fit into the P2' or P3' positions (*Figure 5B and H–J*). Consequently, substrates **10** (KDAGSGGG) and **11** (KDAGGSGG) can still be hydrolysed with relatively high efficiency (ca. 50%) in comparison to PG fragment **2** which contains a pGly bridge (*Figure 5B*). Interestingly, as the P1' position only allows

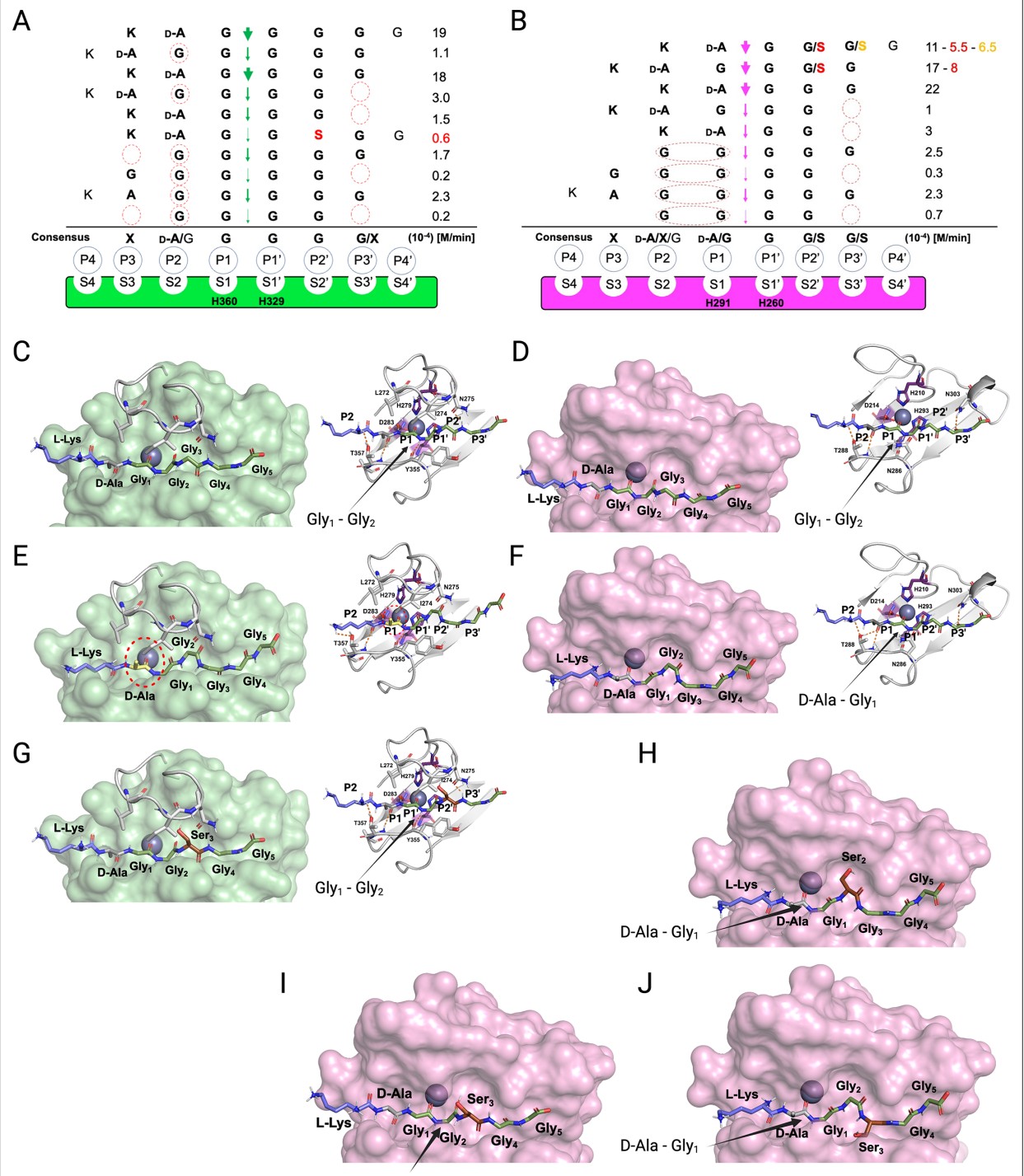

**Figure 5.** Substrate specificity of lysostaphin (LSS) and LytM. Schechter and Berger nomenclature is employed to describe the differences in substrate specificity between LSS (**A**) and LytM (**B**). Scissile bond in the substrate is between the P1 and P1' positions, indicated by green (LSS) and purple arrows (LytM), and hence residues towards the N-terminus from the scissile bond are P1-P4, whereas those towards the C-terminus are designated as P1'-P4'. Peptidoglycan (PG) fragments devoid of stem peptide linked to the C-terminal glycine are shown aligned with respect to their cleavage sites together with the rate of hydrolysis of the particular scissile bond. Consensus sequence displays preferable amino acid(s) that are accepted in the specific position (…P2, P1, P1', P2'…) with respect to the cleavage site. Red circles/ovals indicate missing or less than optimal amino acid accommodation in the particular P site, which translates into reduced catalytic efficiency. Serine substitutions in the glycine bridge and associated rates of hydrolysis are indicated by red and orange colours. (**C–F**) show the docking results for fragment **2** into the catalytic site of LSS and LytM. (**G**) Shows docking result of fragment **11** to LSS. (**H–J**) show the docking results for fragments **10** and **11** into the catalytic site of LytM. LSS and LytM are capable of cleaving the Gly₁-Gly₂ bond in **2** (**C, D**). LytM is also able to cleave the D-Ala-Gly₁ bond (**F**), however, in LSS this would result in a steric clash between the D-Ala sidechain and the residues in loop 1 (**E**). LytM was used at a concentration of 50 µM while LSS was 2 µM. This figure was created using BioRender.com.

Gly, **10** cannot be hydrolysed between $Gly_1$ and $Ser_2$. However, LytM is able to hydrolyse the D-alanyl-glycine bond due to the positioning of the Ser into the P2' position (*Figure 5B and H*). LSS cannot hydrolyse **10** as it would require the disallowed P1' position to accommodate Ser or alternatively P1 to be occupied by D-Ala, both of which are prevented by the extended loop 1 structure in LSS.

Interestingly, when the D-Ala-Gly cross-link is missing, e.g., in pGly or PG fragments **4** and **5**, the latter mimicking a non-cross-linked PG monomer, the cleavage site is shifted within the glycine cross-bridge, i.e., cutting occurs between $Gly_2$ and $Gly_3$ since in minimum a substrate with four residues occupying P2, P1, P1', and P2' sites is required for both LSS and LytM. Owing to this limitation in substrate size and distinct difference in the substrate specificity of LSS and LytM, we observe that while both enzymes can hydrolyse tetraglycine, only LytM can cleave **14** (KDAGG) since it accepts D-Ala in the P1 position. We observed significant reduction in rate of hydrolysis for both LSS and LytM when the substrate is too short, i.e., it cannot fill the P3'/S3' position. This is probably due to the stabilising hydrogen bonds that are formed between the longer glycine chain and a key asparagine residue in loop 4, Asn372 and Asn303 for LSS and LytM, respectively (*Figure 5C and D*).

## Discussion

To gain novel insight into LSS enzyme family specificities, we employed NMR spectroscopy to determine the substrate specificities and the cleavage site using real-time kinetics at atomic resolution on substrates mimicking PG fragments. In addition, we verified by NMR the sites of hydrolysis in muropeptides purified from *S. aureus* USA300 sacculus and compared lytic efficiencies of LSS and LytM towards *S. aureus* USA300 cells with turbidity reduction assays.

Previous research on substrate specificity is largely based on testing several bacterial strains with varying PG composition (*Małecki et al., 2021*; *Ramadurai et al., 1999*) with either lysis assays or purified PG. Thus, determination of the site of hydrolysis often relies on applying abductive reasoning to either include or exclude different PG structures according to the assay results. Because this type of experimental setup often does not reveal the direct cleavage site, nor if there are several cleavage sites it should be supplemented with methods allowing direct observation – such as NMR. Yet, the end-point kinetics type of enzymatic assays, e.g., administering chromatographically purified muropeptides with a lytic catalyst and quenching the reaction after overnight incubation, followed by MS analysis of reaction products, do not necessarily reflect substrate specificity accurately because they only detect reaction products at the end and do not consider plausible secondary reactions.

Our method is superior to the previous approaches because real-time kinetics data combined with a precise determination of the cleavage site reveal the true substrate specificity of LSS. Our data also explain the inconsistencies between results of previous studies, i.e., our extensive set of PG fragments allowed a more comprehensive interpretation than previous studies with fewer substrates (*Bardelang et al., 2009*; *Browder et al., 1965*; *Gründling et al., 2006*; *Małecki et al., 2021*; *Maya-Martinez et al., 2018*; *Reste de Roca et al., 2010*; *Sabala et al., 2014*; *Schneewind et al., 1995*; *Sloan et al., 1977*; *Warfield et al., 2006*; *Xu et al., 1997*). LSS recognises the D-Ala-Gly cross-link. In such a substrate, D-Ala occupies the P2 position, which increases the rate of hydrolysis by 10-fold in comparison to substrates which position glycine into P2 (*Figure 5*). As mature PG in *S. aureus* cell wall is highly (D-Ala-Gly) cross-linked, our results are in excellent agreement with the observed efficiency of LSS towards *S. aureus* cells in numerous studies in the literature (*Kusuma and Kokai-Kun, 2005*; *Małecki et al., 2021*). We also showed that LSS can hydrolyse cell wall in the *S. aureus* ΔfemB mutants (*Strandén et al., 1997*), having three glycines in the cross-bridge, as they contain cross-linked D-Ala-Gly which occupy the P2-P1 positions and two glycines accommodated in the P1'-P2' positions (*Figure 5A*). However, our data also demonstrate that LSS is capable of efficiently hydrolysing glycyl-glycine bonds in PG fragments with different levels of cross-linking, including also non-cross-linked PG monomers. In such a case, $Gly_1$ and $Gly_2$ house the P2 and P1 positions and the scissile bond is shifted from the preferable $Gly_1$-$Gly_2$ bond one residue forward to the $Gly_2$-$Gly_3$ amide bond. This confirms the findings made by Maya-Martinez and colleagues with non-cross-linked PG fragments, i.e., LSS leaves two or three glycines connected to Lys sidechain (*Maya-Martinez et al., 2018*).

Hence, our results provide rationale to the previous observations which show discrepancy regarding the LSS site of hydrolysis. To summarise, LSS prefers cutting between $Gly_1$ and $Gly_2$, whenever $Gly_1$ is cross-linked to D-Ala of neighbouring stem, whereas it hydrolyses the amide bond between $Gly_2$ and $Gly_3$ in non-cross-linked (devoid of D-Ala-Gly bond) PG fragments.

LytM was originally categorised as a glycyl-glycine endopeptidase based on lytic experiments performed using purified cell walls, which showed that it is active against *S. aureus* and *S. carnosus* but not against *Micrococcus luteus* (*Ramadurai et al., 1999*). *M. luteus* PG has the following structure: Ala-(D-γGlu-Gly)-Lys[D-Ala-Lys-D-Glu(-Gly)-Ala-D-Ala-Lys-D-γGlu(-Gly)-Ala]-D-Ala and thus contains neither Gly-Gly nor D-Ala-Gly bonds (*Vollmer et al., 2008*). As was suggested by *Ramadurai et al., 1999* *M. luteus* lacked the bond necessary for LytM cleavage, and here we have identified that bond to be D-Ala-Gly in addition to the known specificity towards glycyl-glycine bonds. D-Ala-Gly bond hydrolysis by LytM was not observed in the recent study on LytM and LSS specificity for PG (*Razew et al., 2023*), which might follow from the type of NMR data acquired. In $^1$H, $^{15}$N correlation spectra product C-terminal glycine amide peaks appear in the 114–117 ppm $^{15}$N region of spectra of samples treated with LytM or LSS (*Razew et al., 2023*). D-Ala-Gly cleavage, however, produces an N-terminal glycine, whose signal is not typically observed in regular N, H correlation spectra due to chemical exchange. We identified all hydrolysis products using $^1$H, $^{13}$C multiple bond correlation NMR spectra acquired from samples dissolved in deuterated buffers. Use of C-H signals is advantageous in that they are not prone to chemical exchange phenomena and enable unambiguous chemical shift assignment.

The only other recognised D-Ala-Gly hydrolase in *S. aureus* is LytN. Contrary to LytM, it contains a catalytic CHAP domain, which cleaves both D-Ala-Gly and MurNAc-L-Ala bonds, in other words is a D-alanyl-glycine endopeptidase as well as an *N*-acetylmuramoyl-L-alanine amidase domain (*Frankel et al., 2011*). D-Ala-Gly cleavage activity has also been found from a CHAP domain of a *S. aureus* Phage ϕ11 murein hydrolase. In this hydrolase amidase activity is contained in another domain of the modular protein (*Sabala et al., 2014*). Recently also *Enterococcus faecalis* EnpA, which belongs to the M23 metalloendopeptidase family, has been shown to cleave D-Ala-Ala bond in *E. faecalis*. D-Ala-Gly cleaving activity was also reported, although without quantitative information regarding catalytic efficiency. Given that the P1 site in EnpA can accommodate D-Ala, it exhibits substrate specificity similar to that of LytM, which allows occupation of the P1 position by D-Ala (and Gly). In this way, LytM and EnpA deviate from LSS which requires invariably glycine in the P1 position. On the other hand, similar to LSS, LytM allows only glycine at the P1' site, whereas EnpA is more promiscuous and can accommodate Gly, L-Ala, and L-Ser in this position (*Małecki et al., 2021*; *Reste de Roca et al., 2010*).

The pivotal findings in this paper are the discovery of the D-Ala-Gly hydrolysis activity of LytM, previously designated solely as a glycyl-glycine endopeptidase. Yet, we show for the first time that the substrate specificity of LSS is defined by the D-Ala-Gly cross-link, which increases catalytic efficiency 10-fold with respect to pentaglycine.

## Materials and methods

### Reagents

Peptides, including the $^{13}$C, $^{15}$N labelled pGly, were synthesised by CASLO as trifluoroacetic acid (TFA) salts. *Escherichia coli* BL21 (DE3)pLysS cells were obtained from Novagen (Novagen). *S. aureus* USA300 (community-acquired MRSA, wild-type strain FPR3757) cells were purchased from American Type Culture Collection (ATCC).

### Protein expression and purification

The catalytic domain of LSS (LSS$_{cat}$ residues 248–384) was produced as a His-tagged GB1-fusion protein. LSS$_{cat}$GB1 was cloned in pET15b vector (Novagen) and heat-shock transformed into *E. coli* BL21(DE3) pLysS (Novagen). The cells were grown at 37°C with 250 rpm orbital shaking in Luria-Bertani broth supplemented with 100 µg/mL ampicillin until the OD$_{600}$ reached 0.55–0.6. Then the temperature was lowered to 20°C and protein expression was induced by adding 0.4 mM isopropyl β-D-1-thiogalactopyranoside. The overexpression of the proteins occurred at 20°C for 20 hr. The cells were harvested by centrifugation at 6000×*g* for 20 min at 20°C followed by cell resuspension in 1× PBS buffer and stored in –80°C until purification. Complete lysis of the cells was achieved using Emulsiflex-C3 homogeniser (Avestin) and lysates were cleared by centrifugation at 35,000×*g* for 30 min at 4°C. The 6x-His-tagged proteins were captured using Ni-NTA Superflow (QIAGEN) and the tag was proteolytically cleaved at 4°C for 16–18 hr. HiLoad 26/60 Superdex 75 pg column (GE Healthcare) was utilised in the size exclusion chromatography (SEC) in 20 mM sodium phosphate buffer pH 6.5, 50 mM

NaCl using an ÄKTA pure system (GE Healthcare). Eluted fractions were concentrated using Amicon Ultra to desired concentration.

LytM catalytic subunits (LytM$_{cat}$ residues 185–316) were cloned into pGEX-2T vector and heat-shock transformed into *E. coli* BL21(DE3)pLysS (Novagen). The GST fusion LytM$_{cat}$ protein was produced the same way as LSS$_{cat}$GB1. The GST fusion proteins were purified with Protino Glutathione Agarose 4B (Macherey-Nagel) and the tag was proteolytically cleaved at 4°C for 16 hr. The protein was further purified by size SEC with a HiLoad 26/60 Superdex 75 pg column (GE Healthcare) in SEC buffer (20 mM sodium phosphate buffer pH 6.5, 50 mM NaCl) using an ÄKTA pure chromatography system (GE Healthcare). Eluted fractions were concentrated using Amicon Ultra to 1 mM final concentration.

The inactive mutant of LytM$_{cat}$ (LytM$_{cat}$_H291A) was generated using QuikChange II Site-Directed Mutagenesis kit (Agilent Technologies) and the mutation was verified by sequencing. Primers used for the mutagenesis were 5'-gggtaattcaacagcgcctgccgtacacttccaacgt-3' (forward primer) and 5'-acgt tggaagtgtacggcaggcgctgttgaattaccc-3' (reverse primer). The recombinant proteins were produced and purified same as the wild type.

The $^{15}$N-labelled LytM$_{cat}$, and LytM$_{cat}$_H291A were expressed in *E. coli* BL21(DE3)pLysS cells in standard M9 minimal medium using 1 g/L $^{15}$N NH$_4$Cl (Cambridge Isotope Laboratories) as a sole nitrogen source, analogously to the protocol described in more detail for the expression of $^{15}$N, $^{13}$C labelled LytM$_{cat}$ (**Tossavainen et al., 2024**). The proteins were purified in 50 mM NaH$_2$PO$_4$, pH 6.5, 50 mM NaCl using the same protocol as described above for the unlabelled proteins.

## *S. aureus* cell growth conditions

*S. aureus* USA 300 cells were grown overnight in Tryptic Soy Broth (TSB) at 37°C, 200 rpm by inoculating one single colony in 3 mL of medium. Bacteria were then diluted 1:100 in prewarmed TSB and grown until desired OD$_{600}$.

## Muropeptide extraction

Muropeptides were extracted as before described with modifications (**Kühner et al., 2014**). 2 mL of bacteria culture at stationary phase were centrifuged at 10,000 rpm in table microcentrifuge, supernatant was discarded, and bacteria were resuspended in 4 mL of 100 mM Tris HCl pH 6.8, 0.25% sodium dodecyl sulphate (SDS) to reach OD$_{600}$ equal to 10. After boiling SDS was removed with extensive washes. Cells were then solubilised with 1 mL of dH$_2$O and incubated for 30 min at room temperature in sonifier water bath. 500 µL of 15 µg/mL DNase in 0.1 M Tris HCl pH 6.8 solution was added and sample was incubated 1 hr at 37°C, 150 rpm. 500 µL of 4 mg/mL Pronase (Sigma) were added and sample was incubated overnight at 37°C, 150 rpm. After enzymes inactivation, the pellet was resuspended with 500 µL of 1 M HCl solution and incubated for 4 hr at 37°C, 150 rpm to release the teichoic acid. The sample was then washed with dH$_2$O until pH was between 5 and 6. Finally, the pellet was resuspended with 12.5 mM sodium dihydrogen phosphate, pH 5.5 or 20 mM sodium phosphate buffer pH 6.5 and adjusted to OD$_{578}$ equal to 3; Mutanolysin (Sigma) solution 5000 U/mL was added and incubated for 16 hr at 37°C, 150 rpm. After mutanolysin inactivation, sample was centrifuged at 10,000 rpm in table microcentrifuge for 10 min, pellet was discarded and muropeptides were in the supernatant. Muropeptides were dried using Savant SC110A SpeedVac (Thermo Fisher Scientific), resuspended with D$_2$O and pH was adjusted to 7.5 using deuterated sodium hydroxide. NMR sample was prepared by diluting the muropeptides stock solution in 50 mM deuterated Tris at pH 7.5, and 0.1 mM DSS as reference compound.

## Sample preparation for NMR kinetics

The synthetic peptides were purchased from CASLO. The peptides were solubilised in D$_2$O at the 20–40 mM concentration, except for Gly$_2$ position $^{13}$C, $^{15}$N labelled pGly which was 6.5 mM. For all peptides the pH was adjusted to 7.5 using deuterated sodium hydroxide. The concentration was determined based on the weight of the dried peptide and the molecular weight taking into consideration the TFA. The exact concentration was verified with NMR using 0.1 mM sodium trimethylsilyl-propanesulfonate (DSS, Chenomx Internal standard, 5 mM 99.9% D, lot PS20190624) as the reference compound. Peptides were diluted to a desired concentration in the presence of 50 mM deuterated Tris pH 7.5. For each sample, prepared to 3 mm OD round-bottom NMR tube, 0.1 mM DSS was added

as the reference. Reactions were initiated by adding the enzyme to a final concentration of 2 µM or 50 µM.

## NMR-based kinetics and resonance assignment of synthetic PG fragments and muropeptides from *S. aureus* sacculus

All NMR experiments were carried out at 25°C, and at the field strength of 800 MHz of $^1H$ frequency on a Bruker Avance III HD NMR spectrometer, equipped with cryogenically cooled $^1H$, $^{13}C$, $^{15}N$ triple-resonance TCI probehead. NMR data collection for kinetics measurements employed a standard $^1H$ pulse program (zgpr) having a selective radiofrequency field for residual HDO signal presaturation during the recycle delay. To ensure quantitative detection of substrate and product concentrations, a 20-s-long recycle delay was used between the transients used for the signal averaging. 0.1 mM DSS was used as a reference compound both for the peak integration, chemical shift referencing, as well as lineshape optimisation. For each time point, experiment was accumulated with 24 transients, yielding an experimental time of 8 min per time point.

First, a reference $^1H$ spectrum was acquired. The sample was then removed from the magnet, and the enzyme was added. The time of enzyme addition was recorded as t=0. After the enzyme addition the sample was placed back into the magnet, and the shim was manually readjusted before the start of acquisition of consecutive $^1H$ spectra to follow the hydrolysis. This preparatory work resulted in a delay of about 5–10 min between enzyme addition (t=0) and the end of the acquisition of the first spectrum. This delay was accounted for in the data analysis for each experiment.

For the resonance assignment of NMR $^1H\alpha$, $^{13}C\alpha$, and $^{13}CO$ chemical shifts in selectively $^{13}C$-labelled $Gly_2$ position in pentaglycine (**1**), a glycine $H\alpha$-detection optimised HCACO-type NMR experiment was devised (*Figure 2—figure supplement 1*). The spectrum was collected as a 2D H(CA)CO $^1H$-$^{13}C$ correlation experiment to establish connectivities between $^1H\alpha$ and $^{13}C'$ resonances in $Gly_2$. The experiment was measured using two transients per FID and the overall experimental time was 200 s per time point.

Assignment of chemical shift resonances in different synthetic PG substrates and products as well as muropeptides from *S. aureus* USA300 sacculus was based on the measurement of $^1H$ as well as 2D $^1H$-$^{13}C$ HSQC, $^1H$-$^{13}C$ HMBC, and $^1H$-$^{13}C'$ selective HMBC spectra. Typically, $^1H$-$^{13}C$ HMBC experiments were measured overnight and at the end of the reaction to warrant the highest sensitivity for product chemical shift assignment.

## Data analysis of NMR kinetics

To obtain substrate and product concentrations at each time point, NMR resonances were integrated together with reference compound using Topspin 3.6.5 software package (Bruker). Rates of reactions were calculated using linear regression of the first 40–60 min of the reaction. Goodness of fitting was evaluated by using the $R^2$ value. All the data fitting had $R^2 \geq 0.9$. A typical experiment was performed with 0.4 mM enzyme and the reaction followed for ~250 min. The slower hydrolysis reactions necessitated relatively long acquisition times of NMR spectra. Therefore, NMR spectrometer stability is of the essence, and well maintained (*Figure 2—figure supplement 3*). To verify reproducibility of the NMR kinetics data, we carried out two independent measurements for PG fragments **2** and **3**. This yielded substrate hydrolysis rates of $2.04\times10^{-3}$ and $1.93\times10^{-3}$ M with, i.e., standard deviation (SD) of $7.778\times10^{-5}$ for **2**, using [LSS]=2 µM. A similar test for **3**, using [LytM]=50 µM, resulted in rates of $9.24\times10^{-4}$ M and $9.35\times10^{-4}$ M, SD $7.778\times10^{-6}$.

## Turbidity reduction assay

Turbidity assay was performed as previously described with some modifications (*Raulinaitis et al., 2017*). *S. aureus* USA 300 bacterial cells were incubated at 37°C, 200 rpm until $OD_{600}$ between 6 and 8 corresponding to late stationary phase. Bacteria were then washed twice using 20 mM Tris HCl pH 7.5, 50 mM NaCl, and resuspended at $OD_{600}$ equal to 5. Bacteria were plated in round-bottom 96-well plate (Thermo Fisher Scientific), and the reaction was started by adding the enzymes at final concentration of 3 µM. The assay was carried out in final volume of 100 µL. Bacteria without any enzyme was used as control sample. Reduction of turbidly of bacteria suspension was followed using BioTek Epoch2 Microplate Spectrophotometer (Agilent Technologies), at 25°C for 16 hr with continuous shaking at 500 rpm. Data were expressed as normalised reduction of the turbidity over the

time. Each experiment was repeated at least in twice and it included quadruplicates. The graphs in *Figure 2D* show the average of the quadruplicates, which have been used for the normalisation and expressed in %.

## Docking

Substrate **2** for docking was built with Maestro molecular modelling software (Schrödinger Release 2021-4: Maestro, Schrödinger, LLC, New York, NY, 2021) and Ligprep ligand preparation tool was used to refine the structure with force field OPLS4. The crystal structures of LytM (PDB code: 4ZYB; *Grabowska et al., 2015*) and LSS (PDB code: 4QPB; *Sabala et al., 2014*) were retrieved from the Protein Data Bank. The protein structures were first prepared with the Protein Preparation Wizard available in the Schrödinger suite. Protein preparation wizard was used to add missing hydrogen atoms, delete water molecules, and assign correct bond orders with force field OPLS4. Glide (*Friesner et al., 2006*; *Friesner et al., 2004*; *Halgren et al., 2004*) was used for docking. The receptor grid was generated with Glide and the Standard Precision (SP) Peptide mode was used for docking. It was noticed that the scoring function favours the strong interaction between the carboxylic acid of the substrate C-terminus and zinc ion, and thus the C-terminus was capped with an *N*-methylacetamide residue. The serine containing peptides were modelled in the binding site by mutating the appropriate residues with Pymol (*Schrödinger and DeLano, 2020*).

## Acknowledgements

We thank Dr. Maarit Hellman for expert technical assistance. This work was supported by the grants from the Academy of Finland and Jane and Aatos Erkko foundation.

## Additional information

### Funding

| Funder | Grant reference number | Author |
| --- | --- | --- |
| Jane ja Aatos Erkon Säätiö | | Perttu Permi |
| Research Council of Finland | 323435 | Perttu Permi |
| Research Council of Finland | 362535 | Perttu Permi |

The funders had no role in study design, data collection and interpretation, or the decision to submit the work for publication.

### Author contributions

Lina Antenucci, Data curation, Formal analysis, Validation, Investigation, Visualization, Writing – review and editing; Salla Virtanen, Validation, Investigation, Visualization, Writing – review and editing; Chandan Thapa, Minne Jartti, Ilona Pitkänen, Investigation, Writing – review and editing; Helena Tossavainen, Conceptualization, Data curation, Formal analysis, Validation, Investigation, Visualization, Writing – review and editing; Perttu Permi, Conceptualization, Resources, Formal analysis, Supervision, Funding acquisition, Validation, Investigation, Methodology, Writing - original draft, Project administration, Writing – review and editing

### Author ORCIDs

Lina Antenucci ⓘ https://orcid.org/0000-0003-1201-9342
Helena Tossavainen ⓘ https://orcid.org/0000-0002-1609-1651
Perttu Permi ⓘ https://orcid.org/0000-0002-6281-1138

Reviewer #1 (Public Review): https://doi.org/10.7554/eLife.93673.3.sa1
Reviewer #2 (Public Review): https://doi.org/10.7554/eLife.93673.3.sa2
Author response https://doi.org/10.7554/eLife.93673.3.sa3

## Additional files

### Supplementary files
- Supplementary file 1. A list of synthetic peptidoglycan (PG) fragments used in this study.
- MDAR checklist

### Data availability
All data generated or analysed during this study are included in the manuscript and supporting files. Source data files have been provided for *Figures 2 and 3*. *Figure 2—source data 1* and *Figure 3—source data 1* contain numerical data used to generate the figures.

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
