## [Editor Report · eLife assessment]

This manuscript describes a **valuable** study aimed at identifying the substrate specificity of two cell wall hydrolases LSS and LytM in *S. aureus*. The authors show that LytM has a novel function of cleaving D-Ala-Gly instead of only Gly-Gly by using synthetic substrates and **compelling** NMR-based real-time kinetics measurements.

---

## [Referee Report · Reviewer #1 (Public Review)]

Summary:

The manuscript aimed at elucidating the substrate specificity of two M23 endopeptidase Lysostaphin (LSS) and LytM in *S. aureus*. Endopeptidases are known to cleave the glycine-bridges of staphylococcal cell wall peptidoglycan (PG). To address this question, various glycine-bridge peptides were synthesized as substrates, the catalytic domain of LSS and LytM were recombinantly expressed and purified, and the reactions were analyzed using solution-state NMR. The major finding is that LytM is not only a Gly-Gly endopeptidase, but also cleaves D-Ala-Gly. Technically, the advantage of using real-time NMR was emphasized in the manuscript. The study explores an interesting aspect of cell wall hydrolases in terms of substrate-level regulation. It potentially identified new enzymatic activity of LytM. However, the biological significance and relevance of the conclusions remain clear, as the results are mostly from synthetic substrates.

Strengths:

The study explores an interesting aspect of cell wall hydrolases in terms of substrate-level regulation. It potentially identified new enzymatic activity of LytM.

Comments on the revised version:

The authors have addressed most of my concerns. I agree that the physiological functions of LytM are not in the scope of the current study.

---

## [Referee Report · Reviewer #2 (Public Review)]

Summary:

This work investigates the enzymatic properties of lysostaphin (LSS) and LytM, two enzymes produced by *Staphylococcus aureus* and previously described as glycyl-glycyl endopeptidases. The authors use synthetic peptide substrates mimicking peptidoglycan fragments to determine the substrate specificity of both enzymes and identify the bonds they cleave.

Strengths:

- This work is addressing a real gap in our knowledge since very little information is available about the substrate specificity of peptidoglycan hydrolases.

- The experimental strategy and its implementation are robust and provide a thorough analysis of LSS and LytM enzymatic activities. The results are very convincing and demonstrate that the enzymatic properties of the model enzymes studied need to be revisited.

- I think the experimental work is extremely well designed and way beyond what people usually do in the field. I believe that this sets a precedent that is worth acknowledging.

---

## [Author Response]

The following is the authors’ response to the original reviews.

**Reviewer #1 (Recommendations For The Authors):**
(1) Figure 2B and 2D: unlike what is written in the results part, the results are not consistent, but opposite: LSS has higher activity in 2B, less in 2D.

The activities in Figure 2B come from NMR kinetic experiments with pGly, whereas Figure 2D reports on activity towards whole *S. aureus* cells. The LytM and LSS activities in these two experiments are indeed not directly comparable, but served to highlight the fact that simple pentaglycine is a poor model substrate for M23 enzymes. We carried out a turbidity assay with pristine enzymatic preparation and indeed it is highly consistent both with the kinetic assay using pentaglycine (Fig. 2B) as well as with larger PG fragments (Fig. 2K) indicating that the catalytic domain of LSS is significantly more efficient than LytM in hydrolyzing cells from community acquired methicillin resistant *S. aureus* strain USA300 as well as synthetic PG fragments. The corresponding paragraph in Results has now been updated and rephrased.

(2) Figure 2, panel K missing statistical analysis, which makes it difficult to appreciate if the difference is significant. If it is a one-time experiment or a single value, the value should be presented as a table. The corresponding text in the results part is confusing. The fold change or drop in percentage is unclear in the figure.

We have added a table (panel L) to Figure 2, which shows absolute values of LSS and LytM hydrolysis rates. Indeed, most of the values are from single NMR kinetic measurements, however, PG fragment (**2**) for LSS and PG fragment (**3**) for LytM were measured as duplicates to verify the reproducibility of the data. This is now mentioned in Figure 3 legend and in the Materials and Methods. Also, the corresponding text in the Results has been updated and rephrased.

(3) Figure 3H: the cleavage of D-ala-gly is unclear, the cleavage products need to be labeled and quantified. The experiment used purified PG treated with mutanolysin. Presumably, mixed monomers, dimers, trimers, and multimers are used. It would be helpful to show the HPLC profile of the purified muropeptide. It would be informative to analyze which fractions generate D-ala-gly. In addition, the intact murein sacculus should be included.

For the sake of clarity, we have moved the 13C-HMBC spectra presented in Figure 3H to Fig. S7 in the Supplementary Material. The full carbonyl carbon region of the reference (prior to addition of enzyme) 13C-HMBC spectrum together with larger expansions of spectra acquired from enzyme-treated muropeptides are now shown. Furthermore, graphical presentations of identified PG fragments due to LSS/LytM activity are included. No HPLC analysis of the muropeptides was performed at this stage. Being insoluble, the intact murein sacculus is not amenable to liquid-state NMR studies, but we envisage studies of this remarkably complex structure also with solid-state NMR.

**Reviewer #2 (Recommendations For The Authors):**
Overall, the experiments address the question asked by the authors and no additional experiments are required to strengthen the conclusions drawn.Abstract:The abstract is not well written and more specific (and accurate) information should be provided by the authors.

We are grateful for the constructive and helpful comments to improve our manuscript. The abstract has now been modified by taking into account the Reviewer’s suggestions.

IntroductionThe intro is relatively long and wordy. It could most certainly be shortened and written in a more simple way to make it more impactful.

The introduction has now been modified by taking into account the Reviewer’s suggestions.

(2) One of the peptide stems in Figure 1 is missing a pentaglycine side chain; I would recommend increasing the font size; the peptide stem looks like it is attached to the carbon in position 2, it may be a good idea to move it to the left?

We thank the Reviewer for this comment. Figure 1 has been improved, the frameshift has been fixed and the non-cross-linked pGly bridge has been included to the lysine side-chain in tetraStem.

ResultsFigure 2 is a bit overwhelming and its description is sketchy. Fig 2B shows a much higher activity of LSS on pGly as compared to LytM whilst 2K shows a very similar rate.

We have rearranged Figures 1 and 2 by moving the original panel J in Figure 2 (structures of PG fragments) to Figure 1 panel C. The bar graph in Figure 2J now shows absolute rates of substrate hydrolysis for 2 mM LSS and LytM. These indicate that LSS is much more efficient against PG fragments in vitro in comparison to LytM. Rates normalized with respect to pGly are shown in Figure 2K. Also, a table showing absolute rates of hydrolysis for 2 mM LSS and 50 mM LytM has been included in Figure 2, panel L. In this Table, the values for PG fragments 2 and 3 were determined by two independent measurements to test and accredit the reproducibility of the method. This is also now elaborated further in the Materials and Methods.

Figure 3 is impressive and very informative but again hard to follow.- Panels 3A and 3B are nicely conceived but the resolution is rather poor and it is difficult to know exactly where the arrows point.

We very much value suggestions given by the Reviewer to improve readability of our manuscript. In the case of Figure 3, we have now greatly enhanced the resolution and readability of the figure by horizontal scaling of panels A and B.

Figure 4 shows a comparative analysis of catalytic rate using various substrates, the authors may want to present graphs with the same y-axis to get the most out of the comparison between substrates.

The scaling of the y-axis is the same for all the substrates now. In addition, we have reorganized the panels in the figure to enhance readability.

Figure 5: - The same remark as above, please cite all panels in alphabetical order.

Citing to Figure 5 has now been revised.

Material and methods:- How were the peptide concentrations determined? It may be useful to indicate if specific conditions were required to solubilize some peptides, pGly is particularly insoluble in aqueous solutions.- Page 19, replace cpm by rpm; biological or technical replicates?

These have now been added and edited accordingly.